# "Thank you for being nice": Investigating Perspectives Towards Social Feedback on Stack Overflow

Mahya Maftouni*  Patrick Marcel Joseph Dubois[†]  Andrea Bunt[‡]

Department of Computer Science, University of Manitoba

## ABSTRACT

The Stack Overflow Q&A community has been frequently criticized for being a harsh, unfriendly environment. Despite numerous calls by the community to improve in this regard, prior work has shown that negative community dynamics continue to deter women, newcomers, and other marginalized groups from getting engaged. Social feedback can play a significant role in shaping community behaviour through group norm reinforcement and can, therefore, be employed as a tool to create more welcoming environments. With this in mind, in this paper we present the design and evaluation of a visible social feedback mechanism for inclusion in a Q&A platform like Stack Overflow. Through an exploratory interview study with 20 Stack Overflow members (10 men, 10 women), we explore users' perceptions of the mechanism's potential benefits and drawbacks. Our findings suggest that compared to the men in our study, the women were more open to additional social feedback on Stack Overflow, finding it a potential solution to make Stack Overflow more welcoming. Our interview findings also suggest that such a tool could be used to encourage newcomers and to allow users to show appreciation for supportive phrasing, complementing Stack Overflow's existing focus on feedback for technically accurate content.

**Keywords:** Q&As, Stack Overflow, social feedback, gender.

**Index Terms:** Human-centered computing—Human computer interaction (HCI)—Empirical studies in HCI; Human-centered computing—Collaborative and social computing—Empirical studies in collaborative and social computing

## 1 INTRODUCTION

Online sharing communities, such as Q&A communities, play an important role in today's knowledge work. They not only serve as key resources for users needing timely technical, problem-solving and troubleshooting advice, but also provide contributors with a platform to showcase their expertise and skills [79]. For example, recruiters will often look at profiles on sites like Stack Overflow to see developers' experience and skills, and connect with them [54]. It is thus critical from an equity standpoint that these communities work for and appeal to all genders, yet prior research shows that this is far from the case [20, 25, 27, 72, 73]. In particular, on Stack Overflow, a popular Q&A platform for software development, less than 10% of members are women [27], despite the fact that, for example, women comprised approximately 24% of computer and information systems professionals in Canada in 2016 [10], with similar numbers in the United States in 2020 [78].

Prior research has uncovered a number of reasons for unbalanced gender representation in online communities. One well-documented factor pertains to community culture [15, 25]. For example, many

---

*e-mail: mahya.maftouni@cs.umanitoba.ca

[†]e-mail:patrick.dubois@cs.umanitoba.ca

[‡]e-mail:bunt@cs.umanitoba.ca

users, but especially women, new coders and other marginalized groups find Stack Overflow's environment hostile [58]. Condescending comments [7] and "boy's club" language [25] on Stack Overflow discourage many people from participating and engaging, but this has been especially true for women [25]. A potential contributor to this hostility is the platform's current perceived emphasis on content accuracy, which is often prioritized over posts that are supportive and encouraging. This raises the question of how more pro-social behaviour might be encouraged and rewarded within a Q&A platform, where current feedback mechanisms (e.g., down/up-votes) tend to reward mainly content accuracy.

In this paper, we take a first step towards a long-term research goal of investigating how interface design might aid in promoting a more welcoming and inclusive atmosphere on Q&A platforms such as Stack Overflow. Mechanisms that elicit and display members' feedback can help increase a community's knowledge of what is valued as well as reinforce intended community norms [1, 35, 45]. With this in mind, we investigate the inclusion of a social feedback mechanism designed to promote pro-social behaviour. Pro-social behaviour refers to "discretionary behaviour such as assisting, comforting, sharing, and cooperating intended to help worthy beneficiaries" [65]. While knowledge-sharing through online platforms such as Stack Overflow is a voluntary action that can benefit the community, here we regard pro-social behaviour as conforming to community values and accepted behaviours (for example, friendliness [60]).

We seek to answer the following research questions: What are users' perceptions of this new feature in terms of its role in a Q&A community like Stack Overflow? What are its potential use cases and envisioned impacts on community dynamics? What are the similarities and differences in how men and women respond to this way of rewarding pro-social behaviour?

To answer these questions, we designed a "Support" feature that allows users to indicate if a post has positive aspects beyond technical accuracy. For example, a user could "Support" a post if it is written with a positive tone or using positive language. They could also "Support" posts from beginners to encourage a more welcoming environment. We propose this "Support" feature as a complement to Stack Overflow's current down/up-vote feedback mechanism. We explore two different ways of rewarding posters who receive "Support"-votes: by having these votes contribute towards reputation points and by ordering posts according to a combination of down/up-votes and "Support"-votes.

To investigate user perceptions of this new "Support" feedback mechanism, we conducted an interview study with 20 Stack Overflow members (10 who self-identified as men and 10 who self-identified as women). Our findings indicate that users saw a range of potential use cases, including encouraging newcomers and recognizing supportive language. Our findings also suggest that the women were more open to having this additional feedback button as compared to men, and found it a potential solution to make Stack Overflow a safer space to post their questions. Like prior research in this area [25, 40], we investigated this issue with an emphasis on the participation of women, comparing their experience to that of men. We acknowledge, however, that gender is not a binary construct and that further research is needed to include the views of non-binary

users.

Our work makes the following contributions: 1) We propose a Support feature for inclusion in a Q&A platform like Stack Overflow that can act as a complement for feedback that emphasizes technical accuracy. 2) We present findings from an interview study that demonstrate users' perceptions of incorporating social feedback in a content-focused Q&A platform. In doing so, we contribute to the literature on gender differences in online communities by highlighting the importance of values embedded in their design. Our findings show the viability of highlighting pro-social behaviour by incorporating peers' social feedback, and open paths to future research on feedback and reward mechanisms that can promote gender diversity.

## 2 RELATED WORK

### 2.1 Gender Differences in Knowledge-Sharing Platforms

Prior research has documented numerous important and nuanced gender differences in online communities. These gender differences manifest themselves in different ways, such as levels of representation [49, 75], types of content contributed [3, 17, 80], content preferences [20, 68], levels of expertise shown [29], levels of confidence conveyed [15, 25] and validation received [20, 38, 70]. There are also examples where women have to engage in additional emotional labour [41] and adopt additional strategies [42] to contribute effectively, whereas the literature suggests that fewer hurdles exist when it comes to men's contributions.

Considering Stack Overflow specifically, studies have shown that the vast majority of contributors are men [72], with women being active for shorter periods of time [73, 82]. This unrepresentative gender balance is becoming a vicious cycle: women show a preference for interacting with other women [8] and they become more active when they encounter other women [24, 46], yet finding women to interact with on the platform is difficult. Further, Stack Overflow users tend to use masculine rather than gender-inclusive language, making women uncomfortable, with many deciding to present themselves as men to fit in [25].

A potential factor in gender participation in online communities is the type of content the community values. As Stack Overflow states in their Help Center, opinion-based questions "don't fit our format well" [59], such questions are often closed by moderators or established users who have the privilege to do so. This topic restriction can be a drawback for using Stack Overflow over other platforms, regardless of gender [79]; however, content analysis of Stack Overflow posts shows that compared to men, women ask more subjective questions that raise discussions and use more tentative language [82]. Similar contrasts have been reported in other online communities [20, 26, 80].

In addition to the types of contributions Stack Overflow encourages, the communication norms seem to penalize women disproportionately. For example, saying "Thank you" is explicitly discouraged on Stack Overflow as comments are "not for socializing" [64]. However, in online interactions, women tend to express their gratitude more frequently than men, and are more concerned about politeness [30]. On Stack Overflow, women post more comments, express their gratitude, and apologize more often than men, and they tend to be more social and use collectivist language in their posts [82]. These communication norms are in contrast to Stack Overflow's more individualistic values [48]. While Stack Overflow treats comments showing appreciation as noise that should be removed, insights from the Stack Overflow Annual Developer survey in recent years have reported that women dislike this policy more than men [43, 55, 56].

These differences in content preferences are highlighted through community validation as well, reinforcing the norms described above. Compared to men, women earn fewer reputation points [40, 72, 82], a virtual reward on Stack Overflow earned through activities such as receiving up-votes. Researchers have proposed a variety of explanations for this difference. May et al. [40] suggest that men's

higher competitiveness compared to women might explain why men thrive on Stack Overflow, as they are more engaged in the "game" to earn reputation points than women [72]. Brooke [8] highlighted gender biases in how answers are scored through down/up-votes on Stack Overflow and questioned the meritocracy in Stack Overflow's scoring system.

Our work contributes by investigating an alternative way to share subjective and socially-oriented feedback on a technically-oriented Q&A, with the aim of supporting norms that more women might find appealing. Stack Overflow's down/up-votes are associated with content accuracy and usefulness, and known to be used by men more often than women [82]. Other feedback mechanisms on social media platforms, like giving "+1" and "Likes" to posts have been used to convey social meaning [67] and appear to have stronger appeal among women compared to men [31]. When Facebook added more reaction buttons, they successfully enabled users to react more precisely to posts, increasing the perceived usefulness of the feedback mechanism [53, 67]. Inspired by these successful implementations of social feedback mechanisms, we therefore explore how adding our own social button-based feedback mechanism can permit Stack Overflow users to communicate more nuanced interpretations of appreciation and social messages, even in a Q&A where the focus is on fast and accurate technical responses.

### 2.2 Influencing Online Behaviours and Norms

Deviations from online community norms and the effects of deviations have been studied widely in different contexts with varying user perceptions and reactions depending on the platform [52] and user gender [21]. With little fear of consequences, some users engage in rude and unwelcome behaviours [28, 39]. Furthermore, as this behaviour can often appear to be normalized, some people tend to perceive it as typical and accepted in these communities, despite harming others' enjoyment and retention [5]. Stack Overflow has been criticized for its unwelcoming environment, with many users expressing concerns over its not only accepted, but enshrined norms [44, 58, 61, 62]. This unwelcoming environment is a deterrent for many users [6, 9, 71, 79]. Prior work further suggests that women see this barrier to engagement as more problematic than men [25].

Given the importance of inclusive knowledge-sharing atmospheres, more research is needed to explore practical approaches to minimizing misbehaviour and promoting inclusion and gender diversity. Prior research on regulating online community behaviour has explored a range of sociotechnical practices [33]. One approach has been to use machine learning techniques, such as classification, to detect online toxicity and negative sentiments [2, 13]. Accurate detection, however, has proven challenging, particularly in light of domain-specific vocabulary [4, 32, 47]. Automated solutions can also contribute to a sense of unfairness since they cannot always consider the context of a post [33]. Others have proposed and studied social approaches, such as involving peers or moderators. For example, League of Legends, a popular online video game, introduced the Tribunal System in 2011, a platform where volunteers could judge a violation reported by other players [36]. In comparison to more automated approaches, involving community members in content moderation can promote a sense of relatedness to the community and care [37]. However, reliance on moderators alone has been shown to be insufficient in creating a welcoming atmosphere considering the frequency of norm violation on certain platforms [13].

Our work adds to this body of knowledge by investigating a preventive approach to online hostility in a Q&A platform such as Stack Overflow. We propose and study the use of visible peer social feedback as a way of helping more community members contribute to shaping norms. In the next section, we provide our rationale for focusing on social feedback and describe how we incorporated this feedback within a prototype Q&A website designed to mimic Stack Overflow.

## 3 INCORPORATING SOCIAL FEEDBACK

Feedback plays a significant role in reinforcing accepted online behaviour [35, 45]. It can consist of "task feedback" and "social feedback" and emphasizing one type of feedback over another can influence a community's interactions and define what is valued [45]. Task feedback relates to the perceived usefulness of the offered post [45]. Stack Overflow enables members to give their feedback on the quality of contributions through down/up-voting. The down/up-votes on Stack Overflow tend to be used as task feedback reflecting the usefulness of the posts. On the other hand, social feedback relates to the behaviour and attributes of the poster [45]. Stack Overflow's Code of Conduct (CoC) advocates friendliness [60], but frequent violations of the Code of Conduct have created a toxic atmosphere [13] and visible social feedback regarding user behaviour is absent.

Considering the critical role of feedback in group norm reinforcement [1] and the importance of social and task feedback balance [45], we are interested in investigating visible social feedback to complement the current sole emphasis on the technical usefulness of posts (i.e., task feedback). Our long-term research objective is to explore how moving beyond the down/up-vote might impact community culture and atmosphere, which could, in turn, potentially encourage more diverse gender participation.

To investigate user perceptions towards social feedback on a content-focus Q&A platform like Stack Overflow, we introduce a Support button as an additional way of reacting to answers and comments in addition to down/up-votes to highlight other important values on a post, for example, language, tone or posters' attitude towards beginners. For this button, we wanted to pick an icon that emphasizes the non-technical nature of this new reaction feature, and that is applicable to different scenarios. After iterating on multiple icon designs and eliciting informal feedback, we found that LinkedIn's support reaction icon, represented by hands holding a heart, best fits our requirements: it depicts the non-technical nature of the Support feature, does not conflict with commonly used icons in other social media, and the hands holding the heart could impart a sense of offering support.

We added a Support button next to each answer and comment. Similar to up-votes, the number of Support-votes received is displayed next to the Support button for comments and on the button for answers and questions (Figure 1).

On Stack Overflow, down/up-votes impact both contributor recognition and content emphasis: down/up-votes contribute to posters' reputation points and change the order in which answers are displayed in a question thread. Correspondingly, to elicit study participant feedback on different ways a new reaction button can highlight and reward pro-social behaviour, we designed two variations of our prototype: a **Points Interface** and an **Order Interface**. In the Points Interface, posters earn one reputation point for each Support-vote they receive. Points earned by Support-votes, Support-points, are displayed in the user profile. They are also shown below their posts inside a heart-shaped icon (Figure 1). The Order Interface orders answers by summing the number of Support-votes and up-votes answers received.

## 4 SEMI-STRUCTURED INTERVIEW EXPLORATORY STUDY

We conducted an exploratory interview study to investigate user perceptions of the new reaction feature that we designed to highlight pro-social behaviour. Our goal for this study was to get initial insights on users' acceptance of this new feature, and their thoughts on how it might impact their participation. We asked participants to interact with a prototype Q&A to ground interview discussions.

### 4.1 Participants

We recruited 10 women and 10 men who are members of Stack Overflow through word-of-mouth and advertising on social media websites (e.g., Reddit). We administered a pre-screening questionnaire that included an open-ended question on gender identity and used responses to this question to recruit an equal number of participants who identified as men and women. Our pre-screening questionnaire and recruitment also welcomed participants who did not identify as either a man or a woman, however, unfortunately we were not able to recruit any non-binary participants. Further research is therefore needed to include the view of non-binary users to explore mechanisms for a fully gender-inclusive online community.

Based on participants' self-reports, nine visit Stack Overflow daily, nine weekly, and two monthly. Five participants were members of Stack Overflow for more than 5 years, seven for 2-5 years, six for 6 months-2 years, and two for less than 6 months (Refer to Table 1 for the gender distribution). Participants received $20 CAD after signing the consent form.

Table 1: Account age and visit frequency of participants by gender

| Account age | Women | Men | Total |
|---|---|---|---|
| more than 5 years | 2 | 3 | 5 |
| 2-5 years | 4 | 3 | 7 |
| 6 months-2 years | 3 | 3 | 6 |
| less than 6 months | 1 | 1 | 2 |
| **Visit frequency** | | | |
| Daily | 5 | 4 | 9 |
| Weekly | 4 | 5 | 9 |
| Monthly | 1 | 1 | 2 |

### 4.2 Design

Our primary focus was on qualitative data from the interviews and qualitative analysis. However, we also included two (between-subjects) study factors to investigate how perceptions might change given i) potential uses of Support-votes within the Q&A platform and ii) gender of participants. The first factor was Interface Type, which had two levels corresponding to our two interfaces: the Points Interface and the Order Interface. Descriptions of these interfaces can be found in Section 3. We used these two interfaces to prompt users to reflect on rewarding and highlighting pro-social behaviour on the platform. Gender was our second between-subjects factor. We assigned participants to an Interface Type randomly, balancing the number of men and women per Interface Type.

### 4.3 Interface and Q&A Content

To explore users' perception and acceptance of the additional reaction button, we implemented a web-based prototype of a Q&A interface that served to prompt participant reflection on how they might use an additional reaction button and how it may affect their engagement with Stack Overflow. To this end, we made the prototype Q&A's layout and appearance (e.g., font family, font size and colours) as similar as possible to Stack Overflow's (see the Supplementary Materials for screenshots). To this prototype, we added the Support button next to each answer and comment as described in the previous section (and shown in Figure 1).

To populate the prototype with ecologically valid data, we collected questions, answers and comments from Stack Overflow's archive using the following popular tags: python, java, c++, css and sql. Since our focus was on the Support feature and not the Q&A content, we selected questions that seemed simple and not too long. We also collected a range of comments from Stack Overflow archival data showing frustration, sarcasm, gratitude, and support. We used a manual process for selecting content for the prototype,

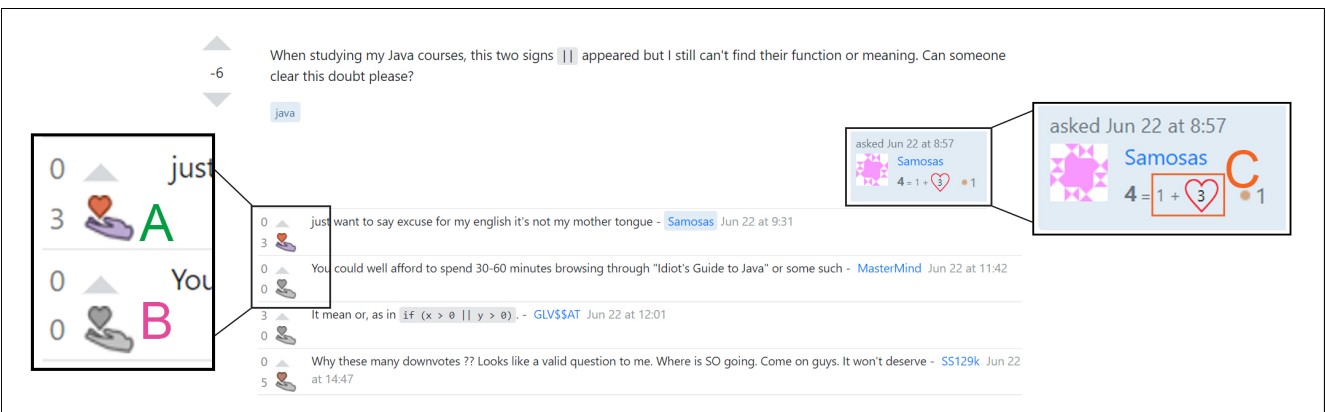

Figure 1: A sample answer in a prototype Stack Overflow interface (referred to the Points Interface in the interview study) - The Support button icon has different colours depending on whether a comment or an answer is Supported or not. (A) A "Supported" comment (B) A comment with no Support-vote (C) The number of regular reputation points and Support-points the user has received.

so our aim was to include enough content that participants could explore the Support feature in a variety of situations while being mindful of resources. We included 8 questions, 14 answers, and 26 comments, which pilot testing suggested provided participants with sufficient content to explore the prototype.

### 4.4 Procedure

Our study sessions were held online due to COVID-19 restrictions and lasted approximately 60 minutes. Each session started with an informal introduction. Participants then had 15 minutes to interact with the prototype, during which time we asked them to read at least 3 question threads, explore user profiles and to use the down/up-vote and Support buttons when they saw fit. Our informal pilot testing suggested that this duration provided sufficient familiarization for the semi-structured interviews, which was our primary data collection method. Since our focus was on community reactions to posted content, we did not ask participants to contribute any new content. Prior to participants interacting with the prototype, a guided tour demonstrated how the Support feature changes the recipient's reputation points in the Points Interface or ranks answers in the Order Interface.

After interacting with the prototype, participants answered a short questionnaire comprised of three Likert-scale questions on their acceptance of the Support feature. Finally, we conducted a semi-structured interview where we asked participants about their attitudes towards the Support feature, how they used the feature while interacting with the prototype, and their views on potential use-cases, benefits, and drawbacks. During the interviews, we also introduced the Interface Type that participants did not experience to elicit preliminary comparative reactions.

### 4.5 Data Collection and Analysis

Our primary source of data was the semi-structured interviews. We also collected participants' interaction data with the prototype (e.g., uses of up-votes, down-votes and Support buttons), and their responses to the post-interaction questionnaire.

To analyze the interviews, we first transcribed them in full. We then created affinity diagrams of participant quotes that captured the participants' experiences with Stack Overflow and their perceptions of the proposed social feedback. The first author, who also conducted the interviews, grouped quotes about similar topics or feelings and developed initial themes. To lessen our own implicit biases, we removed gender tags from participants' quotes during this phase. Then the three authors collaboratively revised the themes, revisiting

the raw data frequently. In doing so, we also looked carefully for any counterexamples to our developed themes.

After developing initial themes, we added the gender labels and looked for gender differences in the collected quotes and themes. Being aware of the complexity associated with gender-based analysis [16] and given our small sample size, we did not expect clear gender distinctions to emerge from our interviews. We uncovered subtle gender differences in our findings that we ground in prior work in order to have a better grasp of the potential benefits of a social feedback feature for men and women.

To analyze the quantitative data, which was not normally distributed, we used the non-parametric Mann-Whitney U Test. We report results as significant if $p \leq 0.05$.

## 5 FINDINGS

### 5.1 Interview Findings

We begin by presenting key themes from our interview data along with supporting quotes. Note that we use man (/woman) to refer to a participant who self-identified as a man (/woman). We use the annotations M and W to denote quotes from men and women, respectively. For most themes, we saw evidence of these perspectives across both the men and the women. We explicitly note any gender differences that we observed.

As is common with qualitative analyses in HCI research, we do not present participant counts with our themes. Due to the semi-structured nature of the interviews, participants might not have expressed an opinion about every theme or might only be in partial agreement with some themes. Counts imply binary agreement or disagreement from every participant, but thematic analysis does not require this level of resolution [14]. We also follow the view that "frequency does not determine value" [14], particularly on a topic of inclusivity, where unique views can be as valuable as common perspectives.

#### 5.1.1 Towards Making Stack Overflow a More Welcoming Space

One of the dominant themes that came from our interviews was the potential impact of having social feedback on creating a welcoming environment on Stack Overflow. Participants described how the Support feature could potentially be used to encourage newcomers, create a safer space and complement community moderation. We describe these perspectives in more detail below.

### Recognizing supportive language and encouraging newcomers

We intentionally avoided providing detailed instructions on the purpose of the Support feature to allow potential use cases to emerge from participants. In the interviews, most participants felt that they understood the intent behind this feature and described cases where they could see themselves using it.

Notably, most of the suggested use cases focused on welcoming and encouraging newcomers either by recognizing supportive language, especially towards newcomers, or by explicitly supporting newcomers whose posts suggest that they lack confidence.

> So I felt that the Support button was a really nice way to say "Oh, actually thank you for being nice". [W06]

> I would Support a question that was exposing the person's insecurities. and I would Support [that] to show them that "it's okay if you don't know this. We all have been there, that's OK." [W04]

Participants could empathize with comments with some levels of self-disclosure, such as when a user expresses that they are new to a specific language or framework, shows lack of confidence, or apologizes for posting a question. They wanted to encourage these types of comments either because someone did the same thing for them when they were newcomers or because they had experienced discouraging reactions on Stack Overflow in the past and understood how intimidating such reactions can be.

While explicitly supporting newcomers was not our original design goal, participants described being aware of Stack Overflow's hostility towards newcomers, and wanting to use the Support feature to welcome and encourage them. Concern about hostility towards new users, which makes them hesitant to contribute [71], is an ongoing issue that has existed since the early days of Stack Overflow. For example, the question "Could we please be a bit nicer to new users?" is currently the fourth-most voted question on Meta Stack Exchange. This question was originally asked in September 2008, less than two months after Meta Stack Exchange went live [44].

### The women found the Support feature as a potential solution to make Stack Overflow a safer space

In our interviews, more women than men seemed open to the Support feature and provided more tangible use cases where they could see potential for benefit. This could suggest gender differences in how men and women are responding to Stack Overflow's communication norms.

While some participants described negative reactions they have seen or personally faced on Stack Overflow towards questions that other users found simple, only women mentioned that these negative reactions deter them from posting.

> I've noticed over the years that sometimes people are not nice and they say "oh this is easy. Why are you asking here?" [...] I would not post a question sometimes and I guess it's because of it. [I'm] a bit afraid of getting weird answers. [W06]

Fear of negative feedback is known to be one of the barriers to women's engagement on Stack Overflow [25]. All the women who expressed hesitation to posting questions saw the Support feature as a potential solution to make Stack Overflow more welcoming to simple questions and beginners, and to help create a more inviting atmosphere by encouraging new and established users to compete to earn Support-vote by being "nice".

> [With the Support feature] Someone like me would be less scared to just write his or her questions there and then be active in the community. So I would look at it as a safer community that way. Because people [would be] competing to be more kind, more polite. [W02]

On the other hand, a few participants, most of whom were men, could not differentiate between Stack Overflow's regular up-votes and the proposed Support-vote.

> I assumed Supporting is monetary. When you have something like vote if the Support is not monetary, then what's [the] point compared to [the] voting system? It's something redundant unless it has a different rewarding mechanism than just votes. [M05]

### Users also want to react to unkindness

Some participants mentioned that they would like a negative version of the Support feature to report mean comments. This urge to do something about toxic behaviour seems to come from personal negative past experiences and disappointment when moderators did not get involved in the way participants hoped. A few participants mentioned that this report should have consequences for the recipients, such as restricting their access to the platform.

> We need to kind of restrict those people who are mean, because they are likely [to] discourage [other users]. And those people who get so many negative points for un-Support [should] be banned for a while or they [should] get a warning. [W07]

There is a flag button on Stack Overflow to report unacceptable behaviour, however, participants mentioned that a mean comment might not necessarily be flag-worthy in light of Stack Overflow's policies. They felt that having a negative version of the Support feature could help them express their opinions without waiting for another moderator to approve their report, which might never happen.

> I use the flag very, very rarely. Only when it's abusive. I haven't flagged things when they're just mean 'cause is that flag worthy? I'm not sure. They're a couple [of] times I've used flags. And actually, people have said "no, you're using it wrong". [M04]

Although Stack Overflow relies on community moderation, including casting votes on the posts or choosing official moderators in a formal election, we saw hints of preference for self-governance and less focus on moderators regarding content moderation. Exploring questions posted under the "declined-flags" tag in Meta Stack Overflow also manifests users' frustration when moderators decline their flag. For example, when a user believed a username with misspelled offensive words ("YuckFou") should not be allowed on Stack Overflow, his flag was declined by the moderators [18]. Considering the subjectivity of what users find inappropriate [4] and the unlikelihood of having a perfect consensus in a large community [77], social feedback can potentially give more of a voice to members.

### 5.1.2 Promoting Community Interactions

From the participants' perspective, one of the contrasts between the Support feature and up-votes was that the Support feature enabled the participants to interact with other members and, therefore, promote a sense of community.

To down/up-vote, participants felt they needed to have appropriate topic expertise. Some participants described down/up-voting an answer or a comment as a responsibility because users, including themselves, rely on votes to choose the correct answer. Stack Overflow's guidelines describe up-votes as indicators of "useful and appropriate" questions and answers [66]. Although each user may

have their own interpretation of what is a useful and appropriate post, participants seemed to internalize Stack Overflow policies favoring factual, informational answers. They mentioned that down/up-voting requires evaluating whether an answer is factually correct, which they felt carries a degree of pressure:

> *I think with up-vote I have to know that the answer works, the solution that's been provided works. So I feel like with the up-vote button there's such a pressure to be an expert in the field that has been discussed. [W05]*

With the Support-vote, participants saw the opportunity for greater community interaction. Even if they could not fully certify an answer's correctness, they welcomed the opportunity to interact with contributors without the risk of violating Stack Overflow's policies. They felt that this type of community interaction could help make Stack Overflow less impersonal, and humanize the community. Impersonal interactions is one of the main barriers that discourage women from participation on Stack Overflow [25]. Irrespective of gender, social interactivity promotes knowledge-sharing on Q&A websites [81] and has a positive effect on the quality of shared knowledge [12].

> *I think that the Support feature is more of an emotional describer versus up-vote [which] is strictly [saying] you provided technical information. [W05]*

> *On platforms like Stack Overflow. You don't interact very much with people. You're just passing by people's comments and answers and questions. And they are just, anonymous boxes with weird designs. I think this [Support feature] adds a human factor to it. [M06]*

Alternatively, a few participants, most of whom were men, were not sure if social community interactions belong on Stack Overflow.

> *Maybe I would comment more [if Stack Overflow implements the Support feature]. And others too. But I don't know if commenting [more] goes well with the purpose of that kind of community. [M01]*

For these participants, the existing interaction norms appear to be working and therefore, they did not see value in a design that seeks to alter these norms.

### 5.1.3 Rewarding Pro-Social Behaviour Without Mixing the Technical Aspect

In our study, we investigated two ways to incorporate Support feedback that mirror the way down/up-votes are currently utilized on the platform. Of the two approaches, most participants liked the idea of awarding reputation points to recipients of Support-votes, however, they wanted the two dimensions of reputation to be separated so that they could distinguish "knowledgeable" from "nice" users. Received up-votes and Support-votes manifest members' warmth (including friendliness and helpfulness) and competence, respectively, which are two universal dimensions of how people characterize others [22]. Participants felt that a clear indication of knowledgeability is essential to assessing the reliability of a user's answers.

> *I think what we're supposed to be using [the] reputation for is to kind of assess how trustworthy this person's answer is. [...] I guess I'd like to be able to tell the difference, is this a person who's technically accurate and knowledgeable and are they nice too? [W03]*

Using Support-votes to influence content emphasis was greeted with much less enthusiasm. Most participants did not want answers ordered based on the summation of the number of up-votes and Support-votes they received. Participants mentioned that they want to see the most accurate answer on the top and they rely on the number of up-votes to choose the answer for their question while exploring Stack Overflow.

> *So basically if the Support button has an emotional aspect attached to it and you're adding up this Support and up-votes together, then we might not necessarily be showing the most appropriate or the strongest answer to the question [on top]. [W08]*

While not the dominant opinion, there were a couple of participants who liked the idea of combining Support-votes with up-votes to highlight answers from "nice" users.

> *If Support could give more attention to those helpful and kind guys, I would definitely prefer to see them. A combined point, based on Support and up-vote [...] those couple of responses there would more appeal to me. [W09]*

Thus, participants were open to the idea of having this type of pro-social behaviour rewarded by the platform, but most did not want to see it influence how answers are presented.

### 5.2 Quantitative Results

#### 5.2.1 Feature Usage

Table 2 shows how both men and women interacted with the prototype. The men down-voted posts and up-voted comments significantly more often than the women. This finding agrees with prior work showing men are more engaged in down/up-voting on Stack Overflow [82]. The remaining differences were not significant, however, this is not surprising given the participants' short exposure to the prototype. Some participants also mentioned that they were simply trying out the Support feature as opposed to expressing their opinions in certain instances. We also tested whether feature usage was different between our two Interface Types (the Points Interface and the Order Interface), but did not find any statistically significant differences.

Table 2: Participants interaction with the prototype - Median (IQR). Bolded values are statistically significant.

|  | Women | Men | *p*-value | U | z |
|---|---|---|---|---|---|
| Supported answers | 2.0 (3.5) | 3.0 (2.75) | 0.136 | 30.5 | -1.491 |
| Supported comments | 2.5 (1.5) | 3.0 (2.25) | 1.0 | 50 | 0.000 |
| **down-voted posts** | **0.0 (0.25)** | **1.5 (2.25)** | **0.049** | **27** | **-1.973** |
| up-voted posts | 4.5 (4) | 6.0 (6) | 0.543 | 42 | -0.608 |
| **up-voted comments** | **0.0 (1.25)** | **1.0 (2.5)** | **0.014** | **19** | **-2.460** |

#### 5.2.2 Questionnaire Responses

As illustrated in Table 3, we did not find any statistically significant differences between the men and women in their responses to the post-interaction questionnaire items. On average, women did respond slightly more positively to the Support feature, however, there was also a lot of variability in the data. Part of the variability is likely owing to the fact that we allowed participants to derive their own meaning to the Support feature, which appeared to impact responses. The interviews provided us with the opportunity to probe further into participants' reactions. There was also no statistically significant difference between responses from participants who interacted with the Points Interface, and those who explored the Order Interface.

Table 3: Median (IQR) Post-interaction questionnaire items

| Item | Range | Women | Men | *p*-value | U | z |
|---|---|---|---|---|---|---|
| I would consider using the Support feature if it is available on Stack Overflow. | 1-7 | 6.5 (4) | 5.0 (3) | 0.534 | 42 | -0.621 |
| If Stack Overflow includes the Support feature, I think the members will use it. | 1-7 | 5.5 (3) | 5.9 (3) | 0.535 | 42 | -0.621 |
| The Stack Overflow community would benefit from the Support feature. | 1-7 | 6.5 (2) | 5.5 (2) | 0.328 | 37 | -0.978 |

# 6 DISCUSSION

Our interview study results indicate that participants could see potential applications for social feedback to encourage newcomers and appreciate supportive language. Women, in particular, found it a possible solution to overcome their hesitation to post their questions, where they currently fear negative reactions from peers. Here, in light of prior work, we discuss how integrating social feedback into Q&A platforms might promote diversity. We also describe promising directions for future research.

## 6.1 Using Social Feedback to Change Social Dynamics to Benefit Women

Stack Overflow's down/up-vote binary, which is associated with technical usefulness and known to be used by men more than women [82], is not expressive enough for highlighting other important values exhibited by a post, including the language, tone or posters' attitudes towards beginners, all of which can be critical for creating a more welcoming atmosphere. Our interviews indicated that social feedback can potentially complement down/up-votes and can be a way to express the values down/up-votes cannot. Women seemed more open to the idea of using social feedback than men, especially since they cannot currently give this feedback according to Stack Overflow's policies [55, 56]. While we have focused our design and study on Stack Overflow, the importance of highlighting values beyond the technical usefulness of shared content likely extends to other online knowledge-sharing communities.

One potential use case for the Support feature highlighted by participants was to show appreciation. While Stack Overflow guidelines explicitly discourage users from saying thank you, women dislike this policy more than men [43, 55, 56]. Participants saw the Support feature as a potential workaround for Stack Overflow's restrictive policies. A recent analysis of a random sample from Stack Overflow archival data shows that women users gave praise and expressed gratitude significantly more often than users who are men [82]. Prior research also suggested that women benefit from expressing their gratitude more than men [34].

We learned that in parallel to our research, Stack Overflow conducted a one-month experiment by adding a "thank-you" icon beside each post to enable users to show their gratitude without leaving a comment, in response to the increasing number of "thanks" comments and to reduce moderators' burden [57]. However, the test of this reaction feature was met with very negative reactions from active members who believed this feature to be a step towards converting Stack Overflow to a social networking site [63]. Our results, on the other hand, suggest that adding social feedback could be perceived to promote social interactivity and that most of our participants welcomed this idea as a way to create a warmer atmosphere. While participants' desire to please the experimenter might have contributed to some of these positive responses [19], the contradiction between our findings and community reaction to the added reaction button also emphasizes the importance of including different members' views instead of focusing on louder voices from established members, who are satisfied with the current dynamics of the community and benefit from the status quo.

Another potential use case of the Support feature mentioned by the participants was giving encouragement to a user who apologizes for asking their question. Since we know that women post apologetic content on Stack Overflow significantly more often than men [82], it is possible that using the Support feature as a reaction to a user apologizing for their posted question could help embrace women and encourage them to engage with the community. On the other hand, publicly displaying the number of Support-votes received might act as a "noob alert" on users' profiles, increasing the risk of their posts not being taken seriously. These potential trade-offs warrant further investigation.

Although these two use cases showcase the potential of advancing towards a more welcoming environment for women, we emphasize the nuanced nature of gender research. We looked for gender differences in our findings and also consulted prior gender research to discuss potential benefits of adding social feedback in a content-focus Q&A platform for women, however, we acknowledge that this way forward might not be suitable for all women and that some might not be interested in social feedback. We are particularly mindful of re-enforcing stereotypes and oversimplifying people [76].

Regardless of gender, through the curb-cut phenomenon, other groups of users, like potentially new programmers, other marginalized groups [58] and even men overall [74], might benefit from adding a social feedback as well. For example, interview participants described how the social feedback can potentially promote community interactions and reward pro-social behaviour. Expressing Support through the added social feedback can also promote a sense of belonging by making the users feel valued and respected [50]. Stack Overflow is known to have a strong individualistic culture, which discourages participation from people who have more collectivist attitudes [48]. Our findings suggest an avenue to promote participation beyond just women, but also from users with more collectivist attitudes.

## 6.2 Incorporating Social Values into Reward Systems

Stack Overflow relies on gamification to motivate contributions, which is the case with other online knowledge-sharing platforms as well (e.g., Quora and Reddit). Stack Overflow's current reputation system rewards users' activities through up-votes. Ideally, game elements should be hard enough to be interesting and easy enough to be feasible without being frustrating [23]. Prior work suggests that it is extremely difficult for some Stack Overflow members to compete with experts and gain reputation points which may disengage them from the game, but in contrast, some expert members of Stack Overflow complain about the lack of interesting and challenging questions to answer because the system rewards common, easier questions [71]. We speculate that game balance is an issue on Stack Overflow where there are not sufficiently challenging tasks for members from different levels of expertise. Having alternative reward systems based on social feedback that focus on other aspects of users' participation, for example their attitude towards new users (e.g., providing answers that are comprehensible by beginners), might encourage a broader range of participants to engage with the platform and create a more welcoming community. Naturally, however, new reward systems will disrupt the current game balance and any direct concrete effects on the platform (such as additional privileges or badges) will need to be carefully investigated through longitudinal studies.

## 6.3 Limitations and Future Work

Our work has demonstrated positive initial reactions to adding social feedback on a Q&A; however, our participants had a short exposure

to the introduced social feedback feature through interacting with a mock interface rather than experiencing a real live community. Given that our findings suggest viability of the idea, further research is required to explore actual behavioural changes and long-term effects of such a feature. Further long-term studies such as field deployments would be necessary to see if and how users adopt the feature and change their behaviours (including seeing if it would lead to higher participation from women), eventually leading to changes in community norms. In addition, while we suspect that our Support feature is fairly unobtrusive and can easily be adapted to be suitable for other platforms, further research is needed to explore how members from other online communities might perceive its usefulness.

Social interactivity plays an important and positive role in women sharing knowledge online [11, 20, 51, 69]. Incorporating click-based social feedback is but one unobtrusive approach that could potentially increase the social interactivity of a platform and make it less impersonal. More work is needed to explore different avenues for increased interactivity, such as creating sub-communities and leveraging personal connections [25]. In shaping these features, it will be important to consider how to balance the needs and perspectives of established and influential community members with those who are experiencing difficulty with the current norms.

Last but not least, we admit incompleteness of our collected data in the sense of missing non-binary users' views. To design gender-inclusive features on knowledge-sharing Q&A platforms, further research regarding non-binary users' behaviour on these platforms, and potential obstacles to their participation is required.

## 7 CONCLUSIONS

In this paper, we investigated how a Support feature on a Q&A site could be used by community members to give social feedback. Our results from interviews with 20 Stack Overflow members suggest that a social feedback feature can potentially play an important role in forming an online community's descriptive norms, by enabling users to show their appreciation, to encourage contributors and to highlighting pro-social behaviour. We also saw that women in particular were more receptive of the social feedback feature, and in light of prior gender research, how it might promote women's engagement with Stack Overflow. Future research can investigate how this and other social feedback mechanisms might influence gender inclusiveness in online communities.

### ACKNOWLEDGMENTS

We want to thank the study participants for contributing their time. We would also thank the Natural Sciences and Engineering Research Council of Canada (NSERC) for funding this research.

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
