# OpenReview forum: ""Thank you for being nice": Investigating Perspectives Towards Social Feedback on Stack Overflow"
_graphicsinterface.org/Graphics_Interface/2022/Conference — GI 2022_

### Official Review · Reviewer_g226 · 2022-04-04
**The ’devil is in the details’ and this paper has made some inroads into those details.**

**Rating:** 7
**Confidence:** 4

**Review:**

This paper investigates using a complimentary feedback system using ‘support’ votes in StackOverflow to determine if this feature would contribute to making the forum more inclusive, especially for women. The authors created a web-based prototype that had the same look and feel as StackOverflow but with the additional “support” button available. They mined StackOverflow for prototypical examples to populate their prototype and had 20 participants (10M/10W) provide feedback. Their results suggest that women are more open to men for the ‘support’ feature and imply that more social support feedback could be useful for Stack Overflow. As well, they found that this feature could help create a more welcoming environment for new comers.

The authors investigate a complex problem space fraught with social complexities in a platform with a majority of men which is commendable. The method used seems appropriate for this preliminary study to show the potential of what a change to the feedback mechanisms could look like and the feeling that users may have towards it. The results are well presented and clear. The analysis looks appropriate for the data and provides helpful insights. While it is unclear how much new is learned that hasn’t been seen in other feedback type investigations, the specifics of Stack Overflow present an important environment where the results provide direction should the community decide to add more inclusive feedback mechanisms.

The main issue I have with the methodology relates to the design of the support feature as it seemed a bit confusing, leading to users having a number of interpretations making it difficult to tell whether users were commenting on the same thing. Even the title of the paper feels a bit unclear about how to interpret ‘support’. That is, “Thank you for being nice’ suggests that the button could be misconstrued as a ‘thank-you’, which is a nice thing to say, but not that the comment itself is necessarily nice versus an indication that the comment it is about is nice. So, you could end up in the amusing situation of nice people in a loop of ‘thank-you’s’; ‘Thank-you for being nice’, [U01] ‘No, thank-you for being nice’[U02]. W04’s comment, “it’s okay if you don’t know this. We all have been there, that’s OK” also makes it unclear whether ‘support’ is an indication that the user thinks the post is nice versus some other associated feeling of support.

This confusion then provides different interpretations of the reward structure - upvotes are for useful answers, but the support button when interpreted as “We’ve all been there, that’s OK” implies t is the person that provides the support that is the ‘nice’’ one while the post they are referring to isn’t necessarily correlated with a ‘nice’ person. Rather, as pointed out in the discussion, the number of supports you get is probably more indicative of how new you are to the platform. So, becomes a flag that the person is new to the platform which could undermine the confidence in that users answers.

So, in the end, I think the paper may have been stronger by exploring the design of the icon to evoke different interpretations before running this study. Then there would be more control over how the user interprets the button leading to a clearer understanding of the implication of the results of the study. So, providing some reflection/pilot data on the icon and why others for sentiments like, “thanks for answer” vs. “I’ve been there too” vs “Nice answer” etc. would be very helpful, including the style of the icon. On that last point, it does seem like style of the icon seems inconsistent with the platform aesthetics and it isn’t clear whether clicking it is giving of a heart (“Here’s a my support to you” or receiving a heart (“I received your kindness”) leading to ambiguity as well.

As for number of participants in the study, it is unclear that the 20 people in the study is sufficient to make claims about diversity in the participants; the lack of gender diversity seems to suggest insufficient recruiting effort was made given this is a study about a particular design idea impacting the gender challenges in Stack Overflow. Further, as the range of reputation plays an important role on Stack Overflow, that would be an important factor to look at rather than only number of years and frequency of use since having an account. That is, there’s a significant difference between someone who goes there for answers regularly and someone who provides many positively scored answers; i.e., the dimensions of frequency/duration of use and reputation are not necessarily correlated, so diversity on reputation would be important.

I do feel that that the research questions are well supported. However, due to the design of the prototype, the question of whether the design itself impacted the perception wasn’t so well addressed; thus, it is unclear that had a different approach to ‘support’ would have ended up with quite different results. Thus, more justification and design methodology for choosing the particular one from the large range of possible designs being studied is important for this paper.

In the end, I feel there are some important insights that were found through this study about the general direction of adding more social interaction cues in Stack Overflow and the complexity of doing so associated with the pragmatics of adding such a feature. The ’devil is in the details’ and this paper has made some inroads into those details.

---

### Official Review · Reviewer_oFPs · 2022-04-07
**The topic is important and the paper is well structured and written. However, the methods used need more explanation and details and authors failed to properly address the last research question.**

**Rating:** 5
**Confidence:** 4

**Review:**

Summary:
The paper proposes a new support feature for inclusion in a Q&A platform Stack Overflow. Authors evaluated the new feature using sumi-structured interviews as well as questionnaires. Results showed that women are more open to additional social feedback on Stack Overflow than men. It was also noted that the new introduced feature might encourage newcomers to the platform, and show appreciation for positive supporting phrases.

Reasons to accept:
- Literature review is well written and authors summarized prior work with relevant sub-fields
- The paper is well structured and written

Reasons to reject:
- I believe that authors did not successfully address the research question 3 regarding the similarities and differences between men and women for a couple of reasons. First, this analysis was conducted on the quantitative findings of the study and given the sample size, I don't believe that the findings are significant. Second, the sample does not represent all genders (no consideration of LGBTQ individuals) which introduces a limitation to answering this question properly.
- It would be beneficial to see the exact questions asked during the questionnaire.
- Education/Employment might be major factors that effect the Support-votes within the Q&A platform on Stack Overflow. This factor was not studied among participants of the study.
- More information/explanation about the used thematic analysis is needed: for example, when generating the themes, did authors used bottom-up or top-down analysis? How many authors contributed to the coding extraction and analysis? etc.
- I think that the differentiation between the regular Stack Overflow up-vote and the proposed support-vote is a valid concern. Specifically, that is was raised by men and this is the highest population on Stack Overflow (only 6.3% are women according to 2018 statistics). If the proposed feature is implemented, it might not be used my most users as many would find it difficult to distinguish between supporting and up-voting a response. Authors are encouraged to think of potential solutions to overcome this challenge in the new proposed interface.

---

### Official Review · Reviewer_YcbR · 2022-04-14
**Nice straight forward study; with some interesting insights**

**Rating:** 7
**Confidence:** 4

**Review:**

This paper presents a study of a proposed interface feature for stackoverlow. Stackoverflow is notoriously unwelcoming to new community members and, as highlighted by the paper, those identifying as female. The paper nicely articulates how and why this large programming community fails to support new and less represented groups, and motivates the need and design of a simple "care". The paper provides some interesting insights. Overall, I think the paper does a good job and will be a useful contribution for those studying online communities. I do have some strong recommendations for revisions if the paper is accepted.

-The specific contribution (how the paper addresses a gap) and the broader insights to HCI should be discussed. In particular the contribution beyond stackoverflow is not considered, how might your results be generalized?

-The title of the paper is confusing: "Thank you for being nice": Investigating Perspectives Towards Social
Feedback on Stack Overflow -> This seems to hide the large and important focus on the experience of different genders. Given the focus on the experience of women vs men, which is perfectly warranted, why is this no clearly in the title? I believe this somehow misrepresents the content of the paper.

- How were quotes extracted in the affinity diagramming analysis?

-Section 6.2 comes out of left field a bit. The conversation around "game elements" is not really clear... I had to search the paper for the single passage that describes "games" in the related work section (which I had completely forgotten about). This paragraph just needs a little more context. The game balancing part also needs some clarification.

These are ordered in level of concern and how they affect the overall contribution, and means that I can less confidently recommend acceptance.

---

### Decision · Program_Chairs · 2022-04-17

Accept